# Steady Squeezing Flow of Magnetohydrodynamics Hybrid Nanofluid Flow Comprising Carbon Nanotube-Ferrous Oxide/Water with Suction/Injection Effect

**DOI:** 10.3390/nano12040660

**Published:** 2022-02-16

**Authors:** Muhammad Sohail Khan, Sun Mei, Nehad Ali Shah, Jae Dong Chung, Aamir Khan, Said Anwar Shah

**Affiliations:** 1School of Mathematical Sciences, Jiangsu University, Zhenjiang 212013, China; sohailkhan8688@gmail.com (M.S.K.); shabnam8688@gmail.com (S.); 2Department of Mechanical Engineering, Sejong University, Seoul 05006, Korea; nehadali199@yahoo.com (N.A.S.); jdchung@sejong.ac.kr (J.D.C.); 3Department of Mathematics and Statistics, University of Haripur, Haripur 22620, KPK, Pakistan; 4Department of Basic Sciences and Islamiat, University of Engineering and Technology Peshawar, Peshawar 25120, KPK, Pakistan; anwarshah@uetpeshawar.edu.pk

**Keywords:** steady, hybrid nanofluid flow, variable magnetic field, parametric continuation method (PCM), BV4C Schemes

## Abstract

The main purpose of the current article is to scrutinize the flow of hybrid nanoliquid (ferrous oxide water and carbon nanotubes) (CNTs + Fe${}_{3}$O${}_{4}$/H${}_{2}$O) in two parallel plates under variable magnetic fields with wall suction/injection. The flow is assumed to be laminar and steady. Under a changeable magnetic field, the flow of a hybrid nanofluid containing nanoparticles Fe${}_{3}$O${}_{4}$ and carbon nanotubes are investigated for mass and heat transmission enhancements. The governing equations of the proposed hybrid nanoliquid model are formulated through highly nonlinear partial differential equations (PDEs) including momentum equation, energy equation, and the magnetic field equation. The proposed model was further reduced to nonlinear ordinary differential equations (ODEs) through similarity transformation. A rigorous numerical scheme in MATLAB known as the parametric continuation method (PCM) has been used for the solution of the reduced form of the proposed method. The numerical outcomes obtained from the solution of the model such as velocity profile, temperature profile, and variable magnetic field are displayed quantitatively by various graphs and tables. In addition, the impact of various emerging parameters of the hybrid nanofluid flow is analyzed regarding flow properties such as variable magnetic field, velocity profile, temperature profile, and nanomaterials volume fraction. The influence of skin friction and Nusselt number are also observed for the flow properties. These types of hybrid nanofluids (CNTs + Fe${}_{3}$O${}_{4}$/H${}_{2}$O) are frequently used in various medical applications. For the validity of the numerical scheme, the proposed model has been solved by another numerical scheme (BVP4C) in MATLAB.

## 1. Introduction

Heat transfer through the flow of fluid on the plate surface or on the surface of a revolving disk is gaining incredible interest from researchers due to its many uses in the aeronautical sciences and engineerings including chemical processes, thermal-energy-producing systems, geothermal industry, gas turbine rotators, medical equipment, rotating machinery, and computer storage. The squeezing flow produces by the motion of the boundaries play a significant role in polymer processing, hydrodynamical machines, lubrication equipment, etc. Due to its wide range of applications in many modern technologies, it can be considered a good source of heat transmission. Researchers have also updated the squeezing flow through the introduction of new ideas known as nanofluids. Nanofluids are widely used in the fields of micromanufacturing, cancer treatment, power generation, microelectronics, microchannels, thermal therapy, drug delivery, and metallurgical sectors. Choi [1] was probably the first person in technology to work on nanofluid for cooling purposes. From his study, he concluded that putting some nanoparticles into the base fluids (oil, water, and blood) make the fluid more efficient for transferring thermal conductivity. Many researchers in this field have used Choi’s idea while working on nanofluids. Shahid et al. [2] numerically analyzed nanofluid by gyrostatic microorganisms in the porous medium on a stretched surface. Turkyilmazoglu [3] analyzed the slip flow pattern theoretically between concentric circular pipes. Xu et al. [4] performed a numerical study of the pulse flow of GOP water nanofluids in a microchannel for heat transmission. Ganji and Dogonchi [5] scrutinized the behavior of time-dependent MHD squeezing flow of nanoliquids between two plates regarding heat transfer using the Cattaneo–Christov heat flux model and thermal radiation. Reddy and Sreedevi [6] analyzed the impact of the chemical reaction and double classification at porous stretching sheets through nanofluid with thermal radiation regarding mass and heat transport. Arrigo et al. [7] studied the behavior of multipurpose hybrid nanofluid using carbon nanotubes (CNTs). Khan et al. [8] observed the behavior of heat transmission on the plate surface using nanofluids formulated by (CNT) nanomaterial. Water and engine oil have been used as the base fluids by Rehman et al. [9] to analyze the behavior of single and multiwall carbon nanotube nanofluids. Flow characteristics with convective heat transfer via Cu-Ag/water hybrid-nanofluid are investigated by Hassan et al. [10]. The numerical and analytical solution of electro-MHD hybrid nanofluid flow in the porous medium with entropy generation has been investigated by Ellahi et al. [11]. Using CNT, Raza et al. [12] studied the development of heat conduction via peristaltic flow in a porous channel in the presence of a magnetic field. Majeed et al. [13] analyzed the heat transfer behavior with dipole effect for magnetite (Fe${}_{3}$O${}_{4}$) nanomaterials that are injected into the following basic fluids, namely refrigerated, water, and kerosene. Hafeez et al. [14] recently studied the flow characteristics at a revolving disk. Using Fortran Code 21, a numerical simulation of non-Newtonian liquid/Al${}_{2}$O${}_{3}$ nanofluids for (0–4) nanosize particles in a two-dimensional square gap with hot and cold lid-driven movement is performed by the Richardson method [15]. Magnetohydrodynamics (MHD) plays a significant role in a fluid motion and has many applications in different technology based on fluid flow. This is the reason most researchers are interested to study the effects of magnetic fields on the motion of the fluid. Regulating the movement of fluid is a basic principle of MHD, such as for the purpose of proper cooling. This approach can also be used in many technologies that involve the improvement of the thermal conductivity and heat transfer rate. One of the leading applications of this type of study is the merging process of the metal in the furnace under the magnetic field. Brain therapy, malignant tumors, blood pressure, and arthritis are well-known uses of magnetic fields. Hsu [16] analyzed the transient Couette flow between two parallel plates regarding heat transfer in the presence of a magnetic field. Siddiqui et al. [17] studied the movement of the MHD fluid flow in order to monitor various diseases through the respiratory tract. A numerical scheme called the Keller box algorithm has been used to solve MHD flow problems at the porous media [18]. Subhani and Nadeem in [19] considered the fluid theory to study the flow of MHD time-dependent hybrid nanofluids at a porous rotating surface. The numerical study of the 3D flow of Casson nanoliquid with chemical reactions on the stretching surface has been performed by Lokesh et al. [20]. The flow properties of Cu-Al${}_{2}$O${}_{3}$/water hybrid nanofluid with Joule heating and MHD on the shrinking/expanding surface has been studied in [21]. The properties of hybrid nanoliquid flow in the presence of a magnetic field on the stretching surface with slip condition has been studied by Tlili et al. [22]. In the current era of technology, the use of hybrid nanoliquids has gained much attention among researchers due to their excellent thermal properties. Hybrid nanofluids increase the rate of heat transfer compared to simple nanofluids and provide better results. Experimental studies of dissolving nanomaterials with volume fraction (1–100) nm have been conducted to improve heat transfer rate and thermal conductivity [23]. An experimental study has been conducted to analyze the temperature and concentration profile of engine oil using small particles of ZnO-MWCNTs [24]. Bovand et al. [25] investigated the properties of aluminium oxide–water nanoliquid flow at a constant temperature in the gap of two long parallel plates. In his proposed model, he also analyzed the thermophoresis power produced by the temperature variation between walls and liquid. The blending of small components into conventional fluids is a popular technique used to improve the thematic properties, which lead to enhancing the drag force [26]. In addition, a recent study of some other types of nanomaterials used in hybrid nanofluids was conducted by Waini et al. [27,28]. It has also been found that 5 percent of the volume of nanomaterials in the principle fluid is more efficient for maximum heat transfer rate. It is concluded from several studies that 5 percent of the volume fraction of nanomaterials in the principle fluid is more efficient for maximum heat transfer rate. In water-based Fe${}_{3}$O${}_{4}$ nanoliquids, 12–15 percent volume fraction has shown a significant influence on the Nusselt number [29,30,31,32,33]. Yahaya et al. [34] examined heat transfer via Cu-Al${}_{2}$O${}_{3}$/H${}_{2}$O hybrid nanofluid on the stretching plate. The ongoing work is filling the research gap in the current literature by studying the incompressible steady flow of hybrid nanoliquid between two parallel porous plates with the variable magnetic field. The problem under consideration has not been researched and is being researched for the first time. According to the author’s knowledge, the current study is a novelty in the field. The authors have also considered several flow properties in the proposed flow model, such as suction/injection, stretching, and nanomaterial volume fractions, so that the effect of different emerging parameters regarding velocity profile, temperature profile, Nusselt number, and skin friction are studied.

## 2. Mathematical Formulation of the Problem

We consider hybrid nanofluids flow in the gap of two horizontal parallel plates separated by a distance of $H\left(t\right)=H\sqrt{1-\alpha t}$, where *H* represent the gap between plates at t=0. For α>0, the two plates are squeezed until they touch at $t=\frac{1}{\alpha }$ and for α<0, the two plates are separated, as depicted in Figure 1. Furthermore, the top plate is moving toward the bottom plate with velocity $v=\frac{dh}{dt}$. Khan et al. [35] in their work used both discs are perfectly conducted. In the present problem, electrical forces are ignored as they are much smaller than magnetic forces. An applied magnetic field is used to generate the induced magnetic field $\left(b1_{x},0,b2_{z}\right)$. Single and multiwalled carbon nanotubes are blended in base fluid water to form a hybrid nanofluid (CNT-Fe${}_{3}$O${}_{4}$/H${}_{2}$O). Cartesian coordinates system is taken at the center of the bottom plate, where the *x*-axis lies along the horizontal axis and the *z*-axis is orthogonal to the plate. The bottom and top plates are kept at a fixed temperature $T_{0}$ and $T_{H}$, respectively. The flow properties of the hybrid nanofluid CNT-Fe${}_{3}$O${}_{4}$/H${}_{2}$O) under consideration are not time-dependent, resulting in more influence of the variable magnetic field. This sort of influence is considered to be due to magnetic characteristics. We have formulated hybrid nanofluid by dissolving the volume fraction of CNT ($\Phi_{2}$ = 0.5) into the originally formulated ferrofluid (Fe${}_{3}$O${}_{4}$/H${}_{2}$O). The mathematical formulation of the aforementioned hybrid nanofluids by continuity, momentum, magnetic field, and energy conservation equations are as follows [35].

Continuity equation:(1)$$\frac{\partial u}{\partial x}+\frac{\partial v}{\partial x}=0,$$

The momentum equations with magnetic effect [35,36,37]:(2)$$\begin{array}{c} u\frac{\partial u}{\partial x}+v\frac{\partial u}{\partial y}=-\frac{1}{\rho_{hnf}}\frac{\partial P}{\partial x}+\frac{\mu_{hnf}}{\rho_{hnf}}\left(\frac{\partial^{2}u}{\partial x^{2}}+\frac{\partial^{2}u}{\partial y^{2}}\right)-\frac{b_{2}\sigma_{hnf}}{\rho_{hnf}}\left(\frac{\partial b_{1}}{\partial y}-\frac{\partial b_{2}}{\partial x}\right) \end{array}$$
(3)$$\begin{array}{c} u\frac{\partial v}{\partial x}+v\frac{\partial v}{\partial y}=-\frac{1}{\rho_{hnf}}\frac{\partial P}{\partial y}+\frac{\mu_{hnf}}{\rho_{hnf}}\left(\frac{\partial^{2}v}{\partial x^{2}}+\frac{\partial^{2}v}{\partial y^{2}}\right)-\frac{b_{1}\sigma_{hnf}}{\rho_{hnf}}\left(\frac{\partial b_{1}}{\partial y}-\frac{\partial b_{2}}{\partial x}\right) \end{array}$$

Maxwell Equations [37]:(4)$$u\frac{\partial b_{2}}{\partial y}+b_{2}\frac{\partial u}{\partial y}-v\frac{\partial b_{1}}{\partial y}-b_{1}\frac{\partial v}{\partial y}+\frac{1}{\sigma_{hnf}\mu_{e}}\left(\frac{\partial^{2}b_{1}}{\partial x^{2}}+\frac{\partial^{2}b_{1}}{\partial y^{2}}\right)=0$$
(5)$$v\frac{\partial b_{1}}{\partial x}+b_{1}\frac{\partial v}{\partial x}-u\frac{\partial b_{2}}{\partial x}-b_{2}\frac{\partial u}{\partial x}+\frac{1}{\sigma_{hnf}\mu_{e}}\left(\frac{\partial^{2}b_{2}}{\partial x^{2}}+\frac{\partial^{2}b_{2}}{\partial y^{2}}\right)=0$$

The energy Equation [36]:(6)$$\begin{array}{cc} u\frac{\partial T}{\partial x}+v\frac{\partial T}{\partial y} & =\frac{\kappa_{hnf}}{{\left(\rho C_{p}\right)}_{hnf}}\left(\frac{\partial^{2}T}{\partial x^{2}}+\frac{\partial^{2}T}{\partial y^{2}}\right)\\ & +\frac{\mu_{hnf}}{{\left(\rho C_{p}\right)}_{hnf}}\left(4{\left(\frac{\partial v}{\partial y}\right)}^{2}+{\left(\frac{\partial u}{\partial y}\right)}^{2}+{\left(\frac{\partial v}{\partial x}\right)}^{2}+2\frac{\partial u}{\partial y}\frac{\partial v}{\partial x}\right) \end{array}$$
where $b_{1},b_{2}$ are the components of the magnetic field, ${\left(\rho C_{p}\right)}_{hnf}$ is the heat capacity of the hybrid nanofluid, *P* is fluid pressure, *T* is the temperature, $\rho_{hnf}$ is fluid density of hybrid nanofluid, $\sigma_{hnf}$ is electrical conductivity of hybrid nanofluid, and $\mu_{hnf}$ is kinematic viscosity of hybrid nanofluid.

Nanofluid are defined as [31,36]:(7)$$\begin{array}{cc} \; & \nu_{hnf}=\frac{\mu_{h}nf}{\rho_{h}nf},\;\frac{\rho_{hnf}}{\rho_{f}}=\left(1-\Phi_{2}\right)\left(\left(1-\Phi_{1}+\frac{\Phi_{1}\rho_{MS}}{\rho_{f}}\right)+\frac{\Phi_{2}\rho_{CNT}}{\rho_{f}}\right),\\ & \frac{\kappa_{hnf}}{\kappa_{bf}}=\frac{\kappa_{CNT}+\left(m-1\right)\kappa_{bf}+\Phi_{2}\left(\kappa_{bf}-\kappa_{CNT}\right)}{\kappa_{CNT}+\left(m-1\right)\kappa_{bf}+\Phi_{2}\left(m-1\right)\left(\kappa_{bf}-\kappa_{CNT}\right)}\\ & \frac{\kappa_{bf}}{\kappa_{f}}=\left(\frac{\kappa_{MS}+\left(m-1\right)\kappa_{f}+\Phi_{1}\left(\kappa_{f}-\kappa_{MS}\right)}{\kappa_{MS}+\left(m-1\right)\kappa_{f}+\Phi_{1}\left(m-1\right)\left(\kappa_{f}-\kappa_{MS}\right)}\right)\;\\ & \;\frac{{\left(\rho Cp\right)}_{hnf}}{{\left(\rho C_{p}\right)}_{f}}=\left(1-\Phi_{2}\right)\left(\left(1-\Phi_{1}+\frac{\Phi_{1}{\left(\rho C_{p}\right)}_{MS}}{{\left(\rho C_{p}\right)}_{f}}\right)+\frac{\Phi_{2}{\left(\rho C_{p}\right)}_{CNT}}{{\left(\rho C_{p}\right)}_{f}}\right),\\ & \frac{\sigma_{hnf}}{\sigma_{f}}=\left(\frac{\sigma_{CNT}+2\sigma_{f}\sigma_{bf}-2\Phi_{2}\left(\sigma_{bf}-\sigma_{CNT}\right)}{\sigma_{CNT}+2\sigma_{f}\sigma_{bf}+\Phi_{2}\left(\sigma_{bf}-\sigma_{CNT}\right)}\right),\;\frac{\mu_{hnf}}{\mu_{f}}=\frac{1}{{\left(1-\Phi_{1}\right)}^{2.5}{\left(1-\Phi_{2}\right)}^{2.5}},\\ & \frac{\sigma_{bf}}{\sigma_{f}}=\left(\frac{\sigma_{MS}+2\sigma_{f}-2\Phi_{1}\left(\sigma_{f}-\sigma_{MS}\right)}{\sigma_{MS}+2\sigma_{f}+\Phi_{2}\left(\sigma_{f}-\sigma_{MS}\right)}\right), \end{array}$$
with $\kappa_{hnf}$ is the thermal conductivity of hybrid nanofluid, $\kappa_{bf}$ is the thermal conductivity of the Fe${}_{3}$O${}_{4}$-nanofuid, and $\Phi_{1}$, $\Phi_{2}$ are the volume fraction of CNTs.

## 3. Boundary Conditions

The boundary conditions are chosen as [36]:(8)$$\begin{array}{cc} \; & u=0,\;v=V_{0},\;b_{1}=ax,\;b_{2}=-aH,\;T=T_{H}\;\;at\;y=h\left(t\right)\\ & u=U_{0}=ax,\;v=0,\;T=T_{0},\;b_{1}=b_{2}=0,\;\;\;at\;y=0 \end{array}$$

The following similarity transformations [36] are considered for the reducing PDEs (Equation 1)–(Equation 6) to ODEs system,
(9)$$\begin{array}{cc} \; & u=axf^{\prime }\left(\eta \right),\;v=-aHf\left(\eta \right),\;b_{1}=axK^{\prime }\left(\eta \right),\;\\ & b_{2}=-aHK\left(\eta \right),\;\eta =\frac{y}{H},\;\theta \left(\eta \right)=\frac{T-T_{H}}{T_{0}-T_{H}},\; \end{array}$$

Therefore, Equation (Equation 1) is satisfied and the remaining Equations (Equation 2)–(Equation 6) transform into the following form
(10)$$\begin{array}{cc} f^{⁗} & =\frac{\rho_{hnf}}{\rho_{f}}\frac{\mu_{f}}{\mu_{hnf}}\left(S\left(f^{\prime }f^{\prime \prime }-ff^{\prime \prime \prime }\right)\right)+M^{2}SRe_{m}{\left(\frac{\sigma_{hnf}}{\sigma_{f}}\right)}^{2}\frac{\mu_{f}}{\mu_{hnf}}\left(fK^{2}\right)\\ & -M^{2}S^{2}Re_{m}^{2}{\left(\frac{\sigma_{hnf}}{\sigma_{f}}\right)}^{3}\frac{\mu_{f}}{\mu_{hnf}}\left(ff^{\prime }K^{2}-f^{2}KK^{\prime }\right), \end{array}$$
(11)$$K^{\prime \prime }=Re_{m}S\frac{\sigma_{hnf}}{\sigma_{f}}\left(f^{\prime }K-fK^{\prime }\right),$$
(12)$$\begin{array}{cc} \theta^{\prime \prime } & =-SPr\frac{{\left(\rho Cp\right)}_{hnf}}{{\left(\rho C_{p}\right)}_{f}}\frac{\kappa_{f}}{\kappa_{hnf}}\left(\theta^{\prime }f\right)-\frac{\mu_{hnf}}{\mu_{f}}\frac{\kappa_{f}}{\kappa_{hnf}}PrEc\left(4\delta f^{\prime 2}+f^{\prime \prime 2}\right), \end{array}$$
and the transform boundary conditions are
(13)$$\begin{array}{cc} \; & f\left(0\right)=0,\;f^{\prime }\left(0\right)=1,\;K\left(0\right)=0,\;\theta \left(0\right)=1,\\ & f\left(1\right)=A,\;f^{\prime }\left(1\right)=0,\;K\left(1\right)=1,\;\theta \left(1\right)=0, \end{array}$$
where A is the suction/injection parameter, $M^{2}=\frac{H^{2}a\sigma_{f}}{\rho_{f}\nu_{f}}$ magnetic parameter, $S=\frac{aH^{2}}{2\nu_{f}}$ squeeze number, $Re_{m}=\sigma_{f}\nu_{f}\mu_{e}$ Rynold’s Magnetic parameter, $Pr=\frac{\nu_{f}{\left(\rho Cp\right)}_{f}}{\kappa_{f}}$ Prandtl number, $Ec=\frac{1}{{\left(Cp\right)}_{f}T_{H}}{\left(ax\right)}^{2}$ Eckert number, and $\delta =\frac{H^{2}}{x^{2}}$.

Required physical parameters are the Nusselt number and skin friction coefficient, which can be defined as,
(14)$$C_{f}=\frac{\mu_{nf}}{\rho_{nf}{\left(\frac{-\alpha l}{2\sqrt{1-\alpha t}}\right)}^{2}}{\left(\frac{\partial u}{\partial z}+\frac{\partial u}{\partial r}\right)}_{z=h(t)},\;\;\;\;N_{u}=-\frac{\kappa_{n}f{\left(\frac{\partial T}{\partial z}\right)}_{z=h(t)}}{k_{f}\left(T_{0}-T_{u}\right)},$$

In case of Equation (Equation 16), we get
(15)$$\begin{array}{cc} \; & \frac{H^{2}{\left(1-\Phi_{1}\right)}^{2.5}{\left(1-\Phi_{2}\right)}^{2.5}\left(\left(1-\Phi_{2}\right)\left(\left(1-\Phi_{1}+\frac{\Phi_{1}\rho_{MS}}{\rho_{f}}\right)+\frac{\Phi_{2}\rho_{CNT}}{\rho_{f}}\right)\right)}{r^{2}}ReC_{f}=f^{\prime \prime }\left(1\right),\\ & -\theta^{\prime }\left(0\right)=\frac{H\sqrt{1-\alpha t}}{\frac{\kappa_{CNT}+\left(m-1\right)\kappa_{bf}+\Phi_{2}\left(\kappa_{bf}-\kappa_{CNT}\right)}{\kappa_{CNT}+\left(m-1\right)\kappa_{bf}+\Phi_{2}\left(m-1\right)\left(\kappa_{bf}-\kappa_{CNT}\right)}}N_{u}. \end{array}$$

## 4. Numerical Solution by PCM

In this section, optimal choices of continuation parameters are made through the algorithm of PCM for the solution of nonlinear Equations (Equation 10)–(Equation 12) with boundary conditions in Equation (Equation 13):First order of ODETo transform the Equations (Equation 10)–(Equation 12) into first order of ODEs, consider the following
(16)$$\begin{array}{cc} \; & f=P_{1},\;f^{\prime }=P_{2},\;f^{\prime \prime }=P_{3},\;f^{\prime \prime \prime }=P_{4}\\ & K=P_{5},\;K^{\prime }=P_{6},\;\theta =P_{7},\;\theta^{\prime }=P_{8} \end{array}$$
putting these transformations in Equations (Equation 10)–(Equation 12), which becomes
(17)$$\begin{array}{cc} P_{4}^{\prime }= & \frac{\rho_{hnf}}{\rho_{f}}\frac{\mu_{f}}{\mu_{hnf}}\left(S\left(P_{2}P_{3}-P_{1}P_{4}\right)\right)+M^{2}SRe_{m}{\left(\frac{\sigma_{hnf}}{\sigma_{f}}\right)}^{2}\frac{\mu_{f}}{\mu_{hnf}}\left(P_{1}P_{5}^{2}\right)\\ & -M^{2}S^{2}Re_{m}^{2}{\left(\frac{\sigma_{hnf}}{\sigma_{f}}\right)}^{3}\frac{\mu_{f}}{\mu_{hnf}}\left(P_{1}P_{2}P_{5}^{2}-P_{1}^{2}P_{5}P_{6}^{\prime }\right), \end{array}$$
(18)$$P_{6}^{\prime }=Re_{m}S\frac{\sigma_{hnf}}{\sigma_{f}}\left(P_{2}P_{5}-P_{1}P_{6}\right),$$
(19)$$\begin{array}{c} P_{8}^{{}^{\prime }}=-SPr\frac{{\left(\rho Cp\right)}_{hnf}}{{\left(\rho C_{p}\right)}_{f}}\frac{\kappa_{f}}{\kappa_{hnf}}\left(P_{1}P_{8}\right)-\frac{\mu_{hnf}}{\mu_{f}}\frac{\kappa_{f}}{\kappa_{hnf}}PrEc\left(4\delta P_{2}^{2}+P_{3}^{2}\right), \end{array}$$
and the boundary conditions become
(20)$$\begin{array}{cc} \; & P_{1}\left(0\right)=0,\;P_{2}\left(0\right)=1,\;P_{1}\left(1\right)=A,\;P_{2}\left(1\right)=0,\\ & P_{5}\left(0\right)=0,\;P_{5}\left(1\right)=1,\;P_{7}\left(0\right)=1,\;P_{7}\left(1\right)=0, \end{array}$$Introducing parameter q, we obtained ODEs in a q-parameter group,To get ODEs in a q-parameter group, we use q-parameter in Equations (Equation 17)–(Equation 19), and, therefore,
(21)$$\begin{array}{cc} P_{4}^{\prime }= & \frac{\rho_{hnf}}{\rho_{f}}\frac{\mu_{f}}{\mu_{hnf}}\left(S\left(P_{2}P_{3}-P_{1}\left(P_{4}-1\right)q\right)\right)+M^{2}SRe_{m}{\left(\frac{\sigma_{hnf}}{\sigma_{f}}\right)}^{2}\frac{\mu_{f}}{\mu_{hnf}}\left(P_{1}P_{5}^{2}\right)\\ & -M^{2}S^{2}Re_{m}^{2}{\left(\frac{\sigma_{hnf}}{\sigma_{f}}\right)}^{3}\frac{\mu_{f}}{\mu_{hnf}}\left(P_{1}P_{2}P_{5}^{2}-P_{1}^{2}P_{5}P_{6}^{\prime }\right), \end{array}$$
(22)$$P_{6}^{\prime }=Re_{m}S\frac{\sigma_{hnf}}{\sigma_{f}}\left(P_{2}P_{5}-P_{1}\left(P_{6}-1\right)q\right)),$$
(23)$$\begin{array}{c} P_{8}^{{}^{\prime }}=-SPr\frac{{\left(\rho Cp\right)}_{hnf}}{{\left(\rho C_{p}\right)}_{f}}\frac{\kappa_{f}}{\kappa_{hnf}}\left(P_{1}\left(P_{8}-1\right)q\right)-\frac{\mu_{hnf}}{\mu_{f}}\frac{\kappa_{f}}{\kappa_{hnf}}PrEc\left(4\delta P_{2}^{2}+P_{3}^{2}\right), \end{array}$$Differentiation by q reaches at the following system w.r.t. the sensitivities to the parameter-qDifferentiating the Equations (Equation 21)–(Equation 23) w.r.t. by *q*
(24)$$d_{1}^{\prime }=h_{1}d_{1}+e_{1}$$
where $h_{1}$ is the coefficient matrix, $e_{1}$ is the remainder, and $d_{1}=\frac{dP_{i}}{d\tau }$, 1≤i≤8.Cauchy Problem
(25)$$d_{1}=y_{1}+a1v_{1},$$
where $y_{1}$, $v_{1}$ are vector functions. By resolving the two Cauchy problems for every component, we then automatically satisfy the ODEs
(26)$$e_{1}+h_{1}\left(a1v_{1}+y_{1}\right)={\left(a1v_{1}+y_{1}\right)}^{{}^{\prime }}$$
and leave the boundary conditions.Using by Numerical SolutionAn absolute scheme has been used for the resolution of the problem
(27)$$\frac{v_{1}^{i+1}-v_{1}^{i}}{▵\eta }=h_{1}v_{1}^{i+1}$$
(28)$$\frac{y^{i+1}-y^{i}}{▵\eta }=h_{1}y^{i+1}+e_{1}$$Taking of the corresponding coefficientsAs given boundaries are usually applied for $P_{i}$, where 1≤i≤8, for the solution of ODEs, we required to apply $d_{2}=0$, which seems to be in matrix form as
(29)$$l_{1}.d_{1}=0\;\mathrm{or}\;l_{1}.\left(a1v_{1}+y_{1}\right)=0$$
where $a1=\frac{-l_{1}.y_{1}}{l_{1}.v_{1}}$.

## 5. Error Analysis

Error analysis is performed to study the reliability of the proposed model solution. The solution of the proposed model by PCM method is validated by BVP4c in MATLAB. In addition, to further support the validity of the solution of the proposed model, the numerical results of several parameters are tabulated (2) and (3) and displayed graphically. Table 1 illustrates the thermophysical properties of water and nanomaterial. Table 2 shows the comparison of the proposed model solution with the solution in [35]. A closed agreement has been found in the two results for different values of S and Φ. Further support for validating the model solution is provided in Table 3 for the numerical values of $f^{\prime \prime }\left(1\right)$ and $-\theta^{\prime }\left(1\right)$. It is evident that the model solution by two methods PCM and BVP4C are correct up to two decimal places.

## 6. Results and Discussions

In this section, we discuss the impact of various emerging parameters of the proposed model quantitatively via different graphs and tables at the velocity and magnetic profile. Hybrid nanoflow flow is observed between two long parallel plates with variable magnetic fields and phenomena of heat transfer. The impact of different key involved parameters, including Prandtl number (Pr), squeezing parameter (S), Eckert number (Ec), magnetic Reynolds number ($Re_{m}$), magnetic parameter (M), and nanomaterials volume fraction ($\Phi_{1}$, $\Phi_{2}$) are studied to analyze the behavior of mass and heat transfer. Table 1 presents the complete sketch of the thermophysical properties of various nanomaterials. The numerical outcomes of two key flow parameters such as skin friction and Nusselt number are shown in Table 2. For the validity of the numerical solution, the proposed model has been solved through two different numerical schemes (BVP4C, PCM) in MATLAB, and their numerical outcomes are displayed in Table 3. The results obtained from the solution of the model reveals that the thermal flow rate of (Fe${}_{3}$O${}_{4}$-SWCNTs-water) and (Fe${}_{3}$O${}_{4}$-MWCNTs-water) rises from 0.8206 percent to 2.5233 percent and 0.9526 percent to 2.8758 percent, respectively, when the volume fraction of nanomaterial increases from 0.01 to 0.03 as depicted in Table 4.

In fact, as the surfaces move, fluids are squeezed in the channels, resulting in an increase in the velocity of the boundary region. Figure 2b demonstrates the impact of magnetic parameter *M* on the transverse velocity. The magnetic parameter is defined as the ratio of fluid flux to the magnetic diffusivity and it plays an important role in determining the diffusion in the magnetic field along streamlines [36]. From the figure, it has been noticed that the transverse velocity is declining due to a rise in the magnetic parameter *M*. In the same way, it has been noticed that the creation of Lorentz force due to magnetic field surging the resistance in the flow region. As the plates move forward, the flow of fluid in the channel wall is suppressed, resulting in increased velocity in the boundary area. Figure 3a illustrates the variational impact of the squeezing parameter *S* on the axial velocity. The squeezing Reynolds number, $S=\frac{\beta l^{2}}{2v_{f}}$, is the ratio of the normal velocity on the upper plate to the kinematic viscosity of the fluid. It is important to note that the large or small values of *S* indicate slow or rapid movement of the lower plate toward the upper plate. The positive values of S indicate that the bottom plate moves away from the top plate, or the distance between the plates is increased, while the negative values of S indicate that the upper plate moves away from the lower plate, or the distance between two plates is decreased. The figure illustrates that the area that lies in the domain 0≤η<0.55 represents the area near to the lower wall and the area that lies in the domain 0.55≤η≤1 represents the area adjacent to the upper wall. It has been noticed that the increase in flow velocity is improving the velocity profile in the domain η<0.55 It has been observed that the fluids pass rapidly through the narrow channel when the surfaces are compressed, in this way, the velocity profile decline in the domain η>0.55, and the fluid encounter additional resistance in the wide channel. The crossflow exists in the centre of the fluid domain. It has been demonstrated in Figure 3b that the velocity profile does not vary at the critical point η=0.55 with the fluctuation in the squeezing parameter and diminish for the magnetic parameter *M*. In addition, for the injection processes, the rising values of the magnetic parameter *M* reduce the $f^{\prime }\left(\eta \right)$.

The impact of two nanomaterials ( CNTs and Fe${}_{3}$O${}_{4}$) with volume fraction ($\Phi_{1}$ and $\Phi_{2}$) have been demonstrated in Figure 4. It reveals that the fluid flow is positively influenced by the increase in volume fraction parameters. The augmenting values of nanomaterial volume fraction $\Phi_{1}$ and $\Phi_{2}$ reduce the boundary layer thickness of the flow, thus increasing the axial velocity $f^{\prime }\left(\eta \right)$ of the fluid flow in the interval η<0.56, and decreasing in the interval η>0.56. It has been observed that the axial velocity profile is cross-flow in the centre of the boundary layer. Figure 5a demonstrates the impact of the squeezing parameter of the magnetic field profile K(η), it reflects that the magnetic profile K(η) is almost linear at S=0.1, and it becomes parabolic at a larger value of *S*, and approach to the maximum value at the middle of the channel. Figure 5b illustrates the impact of $Re_{m}$ on the magnetic field profile K(η), showing that the rising value of the magnetic Reynolds number increments the kinematic viscosity of the fluid flow. A rise in the kinematic viscosity means a fall in fluid density, which increases the magnetic field from the bottom to the upper plate, as displayed in Figure 5b.

Figure 6a is plotted to notice the impact of squeezing parameter *S* on the temperature profile. The incrementing value of the squeezing parameter *S* for the fluid flow produces friction, which generates a certain amount of heat, as well as increases the kinematics energy of the fluid, which also produces heat. As a result, the mean temperature of the hybrid nanofluid flow in the middle of the channel increases. A rise in Eckert number Ec lead to an increase in the thermal flow rate, as displayed in Figure 6b. In fact, due to the increasing values of Eckert number, fluid friction in nanomaterial is produced with greater intensity. In this physical process, kinetic energy is converted into thermal energy, which ultimately assists the increase in temperature profile, and for high values of Eckert number, the profile of θ(η) becomes parabolic and attains the maximum value at the middle of the channel. In addition, increasing the volume fraction of nanomaterials leads to an increment in the density of fluid particles. In this physical process, fluid flow is appreciated and the thermal properties of the fluid flow particles reduce. Therefore, the increasing value of $\Phi_{1}$ lead to diminishing the temperature profile, as portrayed in Figure 7, and at higher values of $\Phi_{1}$, the θ(η) profile becomes parabolic, and approaches to the bottom at the middle of the channel.

## 7. Concluding Remarks

Hybrid fluid flow between two plates in the presence of variable magnetic field is considered using continuity, Navier–Stokes, Maxwell’s, energy, and transport equations. The governing equations of the proposed flow model are converted into highly nonlinear systems of ODEs using similarity transformation. The numerical scheme PCM has been used for the solution of the flow model. The impacts of many involved parameters in the flow model are discussed via different graphs and tables on the velocity profile, temperature profile, and magnetic profile. The main findings of this research work are:As the magnetic field parameter *M* is increasing, the transverse velocity of the hybrid nanofluid is decreasing because of the increase of electromagnetic force.The Lorentz force, which is produced by a magnetic field, has been observed to increase flow resistance.The velocity profile is reduced in the region η>0.55 because the fluid faces more resistance in the wide channel.The increase in the volume fraction of nanoparticles has a favorable impact on fluid flow. If Φ1 and Φ2 are increased, the thickness of the boundary layer is reduced, resulting in an increase in axial velocity $f^{\prime }\left(\eta \right)$ for η<0.56, otherwise it decreases.The fluid friction increases as the Eckert number rises, converting kinetic energy into thermal energy and raising the temperature profile. The Eckert number θ(η) becomes parabolic as the value of θ(η) increases, approaching the maximum value at the center.

## Figures and Tables

**Figure 1 nanomaterials-12-00660-f001:**
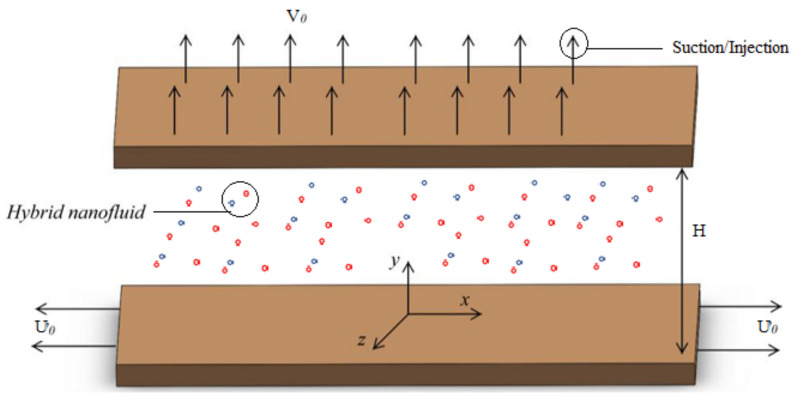
Geometry.

**Figure 2 nanomaterials-12-00660-f002:**
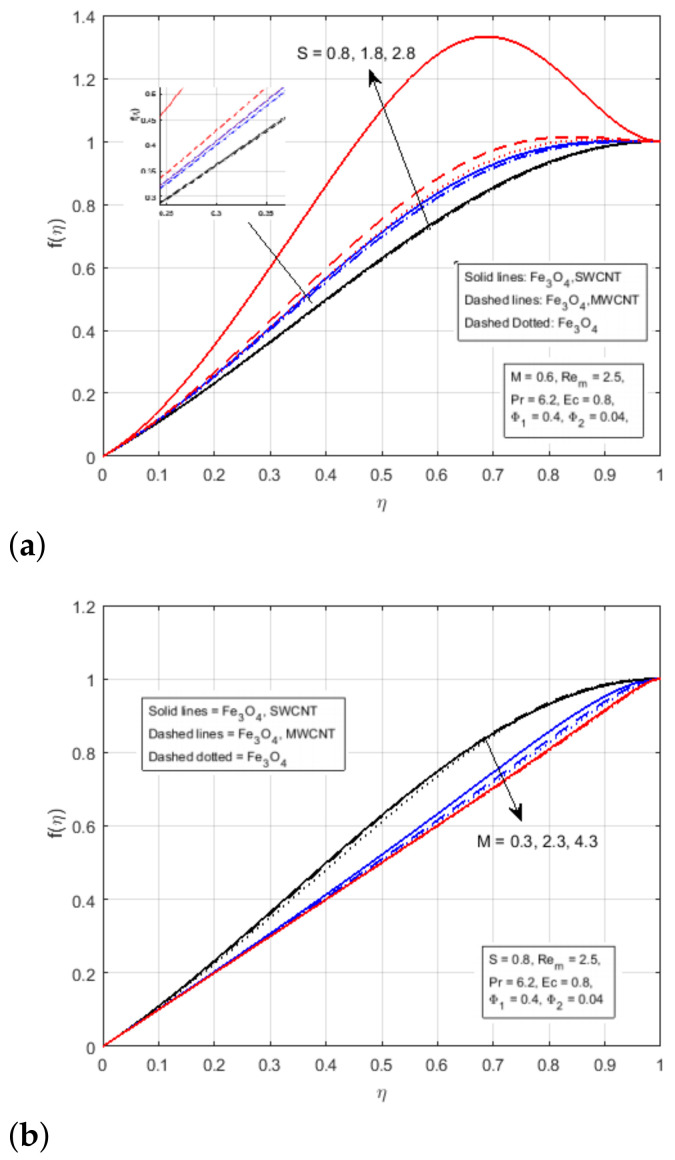
Effect of f(η) for (**a**) *S* and (**b**) *M*.

**Figure 3 nanomaterials-12-00660-f003:**
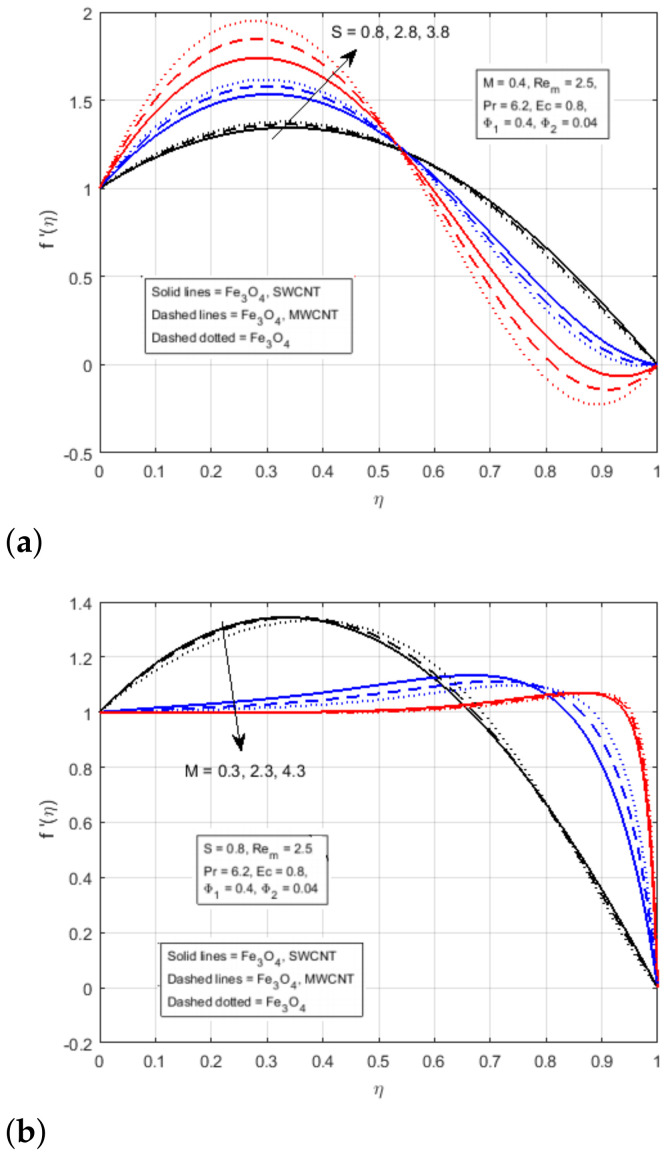
Effect of $f^{\prime }\left(\eta \right)$ for (**a**) *S* and (**b**) *M*.

**Figure 4 nanomaterials-12-00660-f004:**
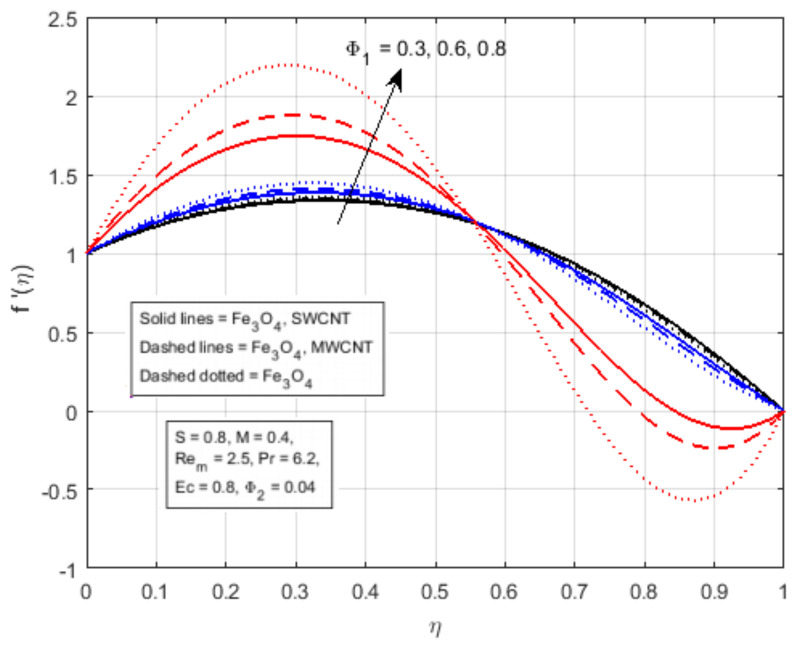
Effect of $f^{\prime }\left(\eta \right)$ for $\Phi_{1}$.

**Figure 5 nanomaterials-12-00660-f005:**
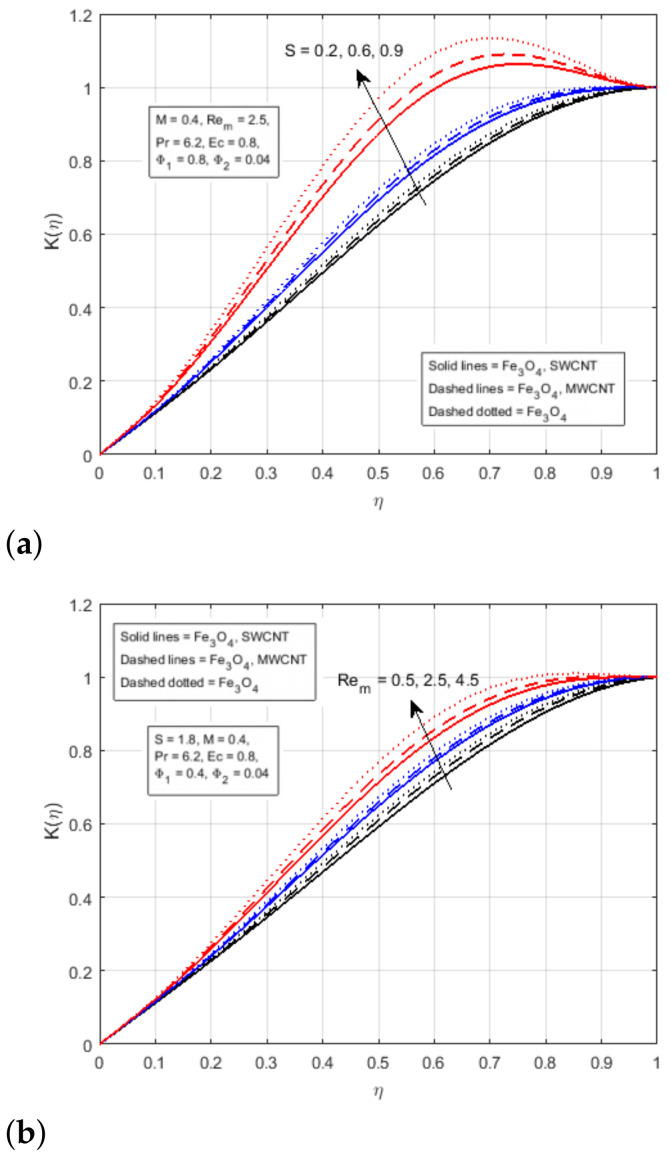
Effect of K(η) for (**a**) *S* and (**b**) $Re_{m}$.

**Figure 6 nanomaterials-12-00660-f006:**
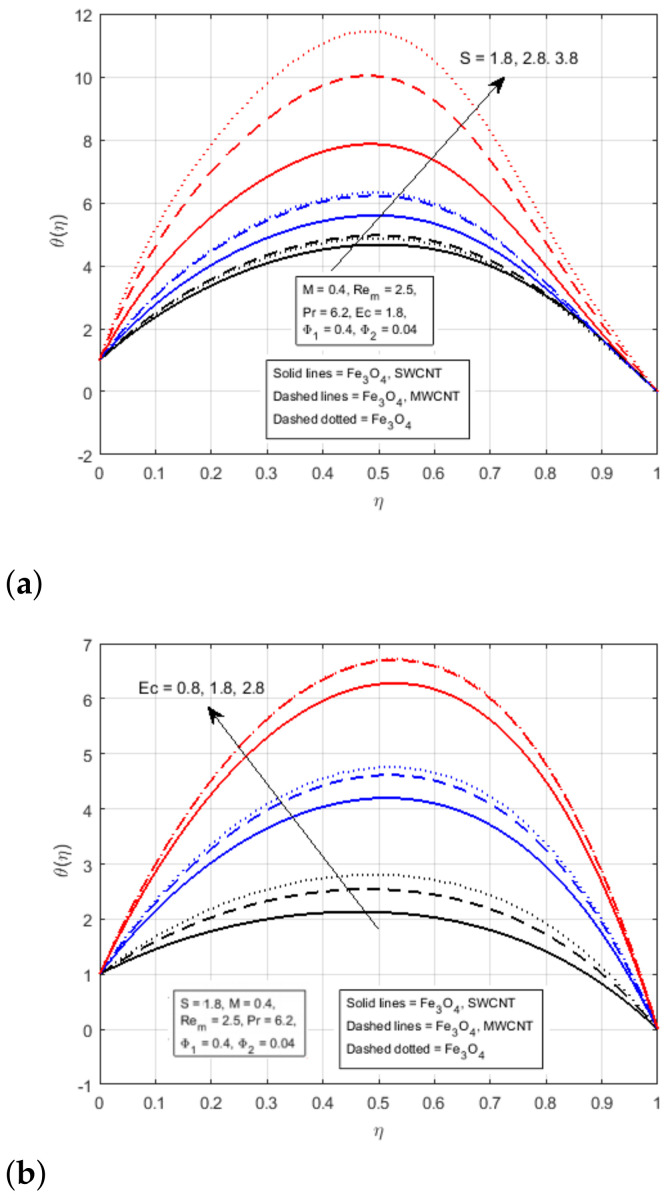
Effect of θ(η) for (**a**) *S* and (**b**) Ec.

**Figure 7 nanomaterials-12-00660-f007:**
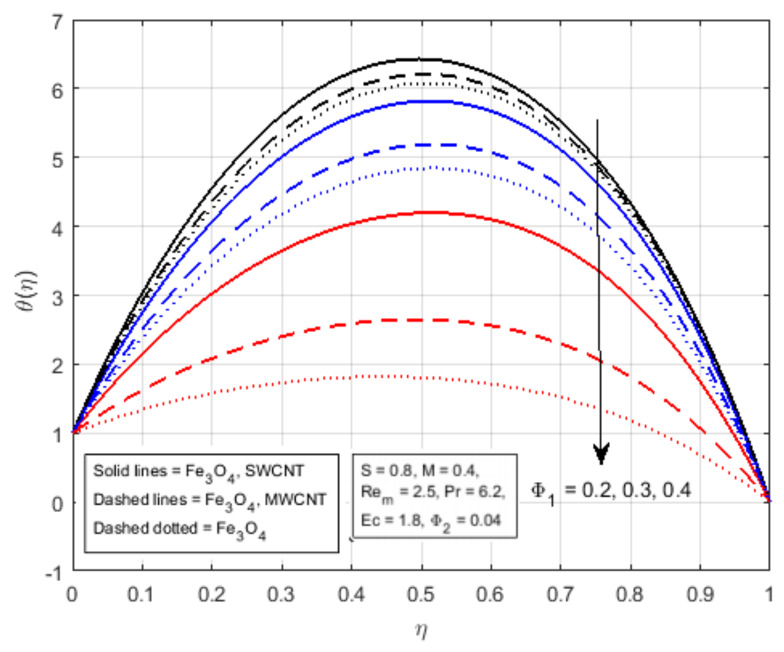
Effect of θ(η) for $\Phi_{1}$.

**Table 1 nanomaterials-12-00660-t001:** The thermophysical properties of water base fluid and hybrid nanoparticles.

	ρ	$C_{p}$	κ	σ
H${}_{2}$O	997.1	4179	0.613	5.5 × $10^{-6}$
Fe${}_{3}$O${}_{4}$	5200	670	6	9.74 × $10^{6}$
SWCNT	2600	425	6600	$10^{6}$
MWCNT	1600	796	3000	$10^{7}$

**Table 2 nanomaterials-12-00660-t002:** Comparison of the numerical results for Nusselt number when Ec = 0.

	*S*	Present	Ali et al. [35]
Φ = 0%	0.1	1.075221	1.078381
	0.5	1.401148	1.403658
	1.0	1.810361	1.813100
	1.5	2.206271	2.201327
Φ = 5%	0.1	1.292331	1.298621
	0.5	1.613601	1.619052
	1.0	2.021148	2.026519
	1.5	2.423006	2.422539
Φ = 10%	0.1	1.572331	1.573849
	0.5	1.883601	1.889474
	1.0	2.291148	2.292857
	1.5	2.693006	2.691972

**Table 3 nanomaterials-12-00660-t003:** Comparison of the numerical results by two methods PCM and BVP4C for skin friction and Nusselt number, with various physical parameters.

			PCM	BVP4C	PCM	BVP4C
$\Phi_{1}$	M	S	$f^{\prime \prime }\left(1\right)$	$f^{\prime \prime }\left(1\right)$	$-\theta^{\prime }\left(1\right)$	$-\theta^{\prime }\left(1\right)$
0.0	0.4	0.2	−3.9734	−3.9704	5.7209	5.7252
0.3			−3.9967	−3.9909	5.7042	5.7038
0.5			−3.9956	−3.9905	0.7386	0.7314
	0.6		−3.9388	−3.9317	0.7410	0.7452
	0.8		−3.8789	−3.8752	0.7434	0.7481
		0.6	−3.6286	−3.6232	0.7397	0.7372
		1.2	−3.8337	−3.8323	0.7082	0.7075

**Table 4 nanomaterials-12-00660-t004:** The heat transfer has been calculated percent wise as for the various nanoparticles Pr = 6.2, S = 1.8, Ec = 0.8, using the percentage formula %increase = $\frac{With\;Nanoparticle}{Without\;Nanoparticle}$× 100 = Result, Result-100 = %enhancment.

$\Phi_{1},\;\Phi_{2}$	$-\theta^{\prime }\left(1\right)\;\mathrm{for}\;{\mathrm{Fe}}_{3}O_{4},\mathrm{SWCNT}$	$-\theta^{\prime }\left(1\right)\;\mathrm{for}\;{\mathrm{Fe}}_{3}O_{4},\mathrm{MWCNT}$
0.0	5.5324	5.5324
0.01	5.5778	5.5851
	(0.8206% increase)	(0.9526% increase)
0.02	5.6242	5.6381
	(1.6593% increase)	(1.9106% increase)
0.03	5.6720	5.6915
	(2.5233% increase)	(2.8758% increase)

## Data Availability

Not applicable.

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
