# Peer review of "Steady Squeezing Flow of Magnetohydrodynamics Hybrid Nanofluid Flow Comprising Carbon Nanotube-Ferrous Oxide/Water with Suction/Injection Effect"

_nanomaterials, 2022, doi:10.3390/nano12040660_

Round 1
Reviewer 1 Report
- I found many minor grammatical errors throughout the manuscript. E.g. “Ganji and Dogonchi.[5] have been investigated…” should be “Ganji and Dogonchi.[5] have investigated…” Please check it carefully.
- The introduction section is poorly written, with sentences and paragraphs disconnected from a coherent description of the state of the art.
- I think the manuscript, even though appears to have a strong mathematical background, lacks of description and connection between the sections of the discussion. It is very hard to follow and understand the relationship between the equations and the presented plots. Also, the mathematical description seems to be dependent strongly on a couple of references, which makes its novelty questionable.
Author Response
TO
Editor-in-Chief
Nanomaterials
Manuscript ID: Nanomaterials-1545970 30 January 2022
Review submission regarding paper title:
Steady Squeezing Flow of Magnetohydrodynamics Hybrid Nanofluid Flow comprising Carbon Nanotube-Ferrousoxide/Water with Suction/Injection Effect
Authors: Muhammad Sohail Khan, Sun Mei, Shabnam, Nehad Ali Shah, Jae Dong Chung, Aamir Khan , Said Anwar Shah
- Department School of Mathematical Sciences, Jiangsu University, Zhenjiang 212013, Jiangsu, China.
- Department of Mechanical Engineering, Sejong University, Seoul 05006, Korea.
- Department of Mathematics and Statistics, University of Haripur, 22620, KPK, Pakistan.
- Department of Basic Sciences and Islamiat, University of Engineering and Technology Peshawar, Peshawar, KPK, Pakistan.
************************************************************************
REPLY TO REVIEWER 1
Q1. I found many minor grammatical errors throughout the manuscript. E.g. “Ganji and Dogonchi. [5] have been investigated…” should be “Ganji and Dogonchi.[5] have investigated…” Please check it carefully.
REPLY: The introduction section has been revised and all the grammatical and typo errors have been corrected. (See with colored red on pages 2-4).
Q2. The introduction section is poorly written, with sentences and paragraphs disconnected from a coherent description of the state of the art.
REPLY: The introduction section has been rewritten with the proper connection between the sentences.
Q3. I think the manuscript, even though appears to have a strong mathematical background, lacks of description and connection between the sections of the discussion. It is very hard to follow and understand the relationship between the equations and the presented plots. Also, the mathematical description seems to be dependent strongly on a couple of references, which makes its novelty questionable.
REPLY: The FORMULATION section has been modified to clear the description of the problem. Also, the discussion section has been revised to clarify the relationship between the equations, tables, and graphs. The novelty of the problem has also been added at the end of the introduction section. (See with colored red on pages 4, 11).

Reviewer 2 Report
To make the manuscript self-contained and more readable, make sure all the variables have been fully specified particularly the physically meaning after each equation.
It is recommended to distinguish the novelty of the paper from the original literatures. I.e., highlight the content contributed by the author by using proper subtitles.
It is recommended to add a theory part for the analysis.
More discussions and literatures should be added in the introduction.
Is there any other limitation of the proposed idea? Such as the specific or limited ranges of working environment or conditions? Is the proposed architecture still valid under other environment?
Carefully recheck grammar and typo errors.
Author Response
TO
Editor-in-Chief
Nanomaterials
Manuscript ID: Nanomaterials-1545970 30 January 2022
Review submission regarding paper title:
Steady Squeezing Flow of Magnetohydrodynamics Hybrid Nanofluid Flow comprising Carbon Nanotube-Ferrousoxide/Water with Suction/Injection Effect
Authors: Muhammad Sohail Khan, Sun Mei, Shabnam, Nehad Ali Shah, Jae Dong Chung, Aamir Khan , Said Anwar Shah
- Department School of Mathematical Sciences, Jiangsu University, Zhenjiang 212013, Jiangsu, China.
- Department of Mechanical Engineering, Sejong University, Seoul 05006, Korea.
- Department of Mathematics and Statistics, University of Haripur, 22620, KPK, Pakistan.
- Department of Basic Sciences and Islamiat, University of Engineering and Technology Peshawar, Peshawar, KPK, Pakistan.
************************************************************************
REPLY TO REVIEWER 2
To make the manuscript self-contained and more readable, make sure all the variables have been fully specified particularly the physically meaning after each equation.
Q1: It is recommended to distinguish the novelty of the paper from the original literatures. I.e., highlight the content contributed by the author by using proper subtitles.
REPLY: Novelty of the problem has been added at the end of the introduction section. (See with colored red on page 4).
Q2: It is recommended to add a theory part for the analysis.
REPLY: The error analysis section has been added to the paper. (See with colored red on page 9).
Q3: More discussions and literature should be added to the introduction.
REPLY: Both the introduction and Discussions sections have been revised. The most recent research papers have been included. Also, the discussion section has been modified. New graphs have been added for a clear picture of the results. (See colored red on pages 3, 4, 10).
Q4: Is there any other limitation of the proposed idea? Such as the specific or limited ranges of working environment or conditions? Is the proposed architecture still valid under another environment?
REPLY: The current working environment or conditions were selected as these working environments have controlled the system error. Any working environment could be selecteddepending on the system error. If the error is in control, so, another environment could be selected.
Q5: Carefully recheck grammar and typo errors.
REPLY: All the grammar and typo errors have been corrected according to the best of our knowledge.

Reviewer 3 Report
The following points must be addressed before publication:
- The abstract must be improved. The Comparative-quantitative results should be mentioned in the Abstract.
- Please explain in detail what makes your study different from the available literature.
- It would be better for the authors to arrange the summary of each section following the sequence.
- More Comparative-quantitative results should be mentioned in the Conclusion.
- It is highly recommended to improve the English quality of the manuscript. There are a lot of grammatical errors.
Author Response
TO
Editor-in-Chief
Nanomaterials
Manuscript ID: Nanomaterials-1545970 30 January 2022
Review submission regarding paper title:
Steady Squeezing Flow of Magnetohydrodynamics Hybrid Nanofluid Flow comprising Carbon Nanotube-Ferrousoxide/Water with Suction/Injection Effect
Authors: Muhammad Sohail Khan, Sun Mei, Shabnam, Nehad Ali Shah, Jae Dong Chung, Aamir Khan, Said Anwar Shah
- Department School of Mathematical Sciences, Jiangsu University, Zhenjiang 212013, Jiangsu, China.
- Department of Mechanical Engineering, Sejong University, Seoul 05006, Korea.
- Department of Mathematics and Statistics, University of Haripur, 22620, KPK, Pakistan.
- Department of Basic Sciences and Islamiat, University of Engineering and Technology Peshawar, Peshawar, KPK, Pakistan.
************************************************************************
REPLY TO REVIEWER 3
The following points must be addressed before publication:
Q1. The abstract must be improved. The Comparative-quantitative results should be mentioned in the Abstract.
REPLY: The abstract and conclusion section have been revised. Also, some of the obtained results have been mentioned in the abstract section. (See colored red on pages 1, 15).
Q2. Please explain in detail what makes your study different from the available literature.
REPLY: Novelty of the paper has been added at the end of the introduction section. (See colored red on page 4).
Q3. It would be better for the authors to arrange the summary of each section following the sequence.
REPLY: Summary of the problem has been added in the abstract and conclusion sections. (See with colored red on pages # 1, 15)
Q4. More Comparative-quantitative results should be mentioned in the Conclusion.
REPLY: The Conclusion section has been modified and comparative-quantitative results are mentioned in the modified section. (See with colored red on pages # 15)
Q5. It is highly recommended to improve the English quality of the manuscript. There are a lot of grammatical errors.
REPLY: The English quality has been improved to the best of our knowledge. Any other suggestion for its improvement is welcomed. Thank You

Round 2
Reviewer 1 Report
- I still found several grammatical errors, and even repeated phrases within the same paragraph. Not all the acronyms are explained at the first mention within the text.
- The authors did not review the coherence between sections and pharagraphs, and only limited to add more text in the same fashion as the existing text.
- What is κMS? why you have Φ in uppercase and lowercase indistinctively? is there any difference? Please check all the variables and coefficients are properly described!
- The only section which has improved with respect to the original is section 6 "Results and Discussions". Sections 1-5 need to be thoroughly revised.
Author Response
TO
Editor-in-Chief
Nanomaterials
Manuscript ID: Nanomaterials-1545970 5 January 2022
Review submission regarding paper title:
Steady Squeezing Flow of Magnetohydrodynamics Hybrid Nanofluid Flow comprising Carbon Nanotube-Ferrousoxide/Water with Suction/Injection Effect
Authors: Muhammad Sohail Khan, Sun Mei, Shabnam, Nehad Ali Shah, Jae Dong Chung, Aamir Khan , Said Anwar Shah
- Department School of Mathematical Sciences, Jiangsu University, Zhenjiang 212013, Jiangsu, China.
- Department of Mechanical Engineering, Sejong University, Seoul 05006, Korea.
- Department of Mathematics and Statistics, University of Haripur, 22620, KPK, Pakistan.
- Department of Basic Sciences and Islamiat, University of Engineering and Technology Peshawar, Peshawar, KPK, Pakistan.
************************************************************************
REPLY TO REVIEWER 1
Comments and Suggestions for Authors
Q1. I still found several grammatical errors, and even repeated phrases within the same paragraph. Not all the acronyms are explained at the first mention within the text.
Ans. All the grammatical mistakes are corrected to the best of our knowledge.
Q2. The authors did not review the coherence between sections and pharagraphs, and only limited to add more text in the same fashion as the existing text.
Ans. The authors tried their best to make coherence between sections.
Q3. What isκMS? why you have Φ in uppercase and lowercase indistinctively? is there any difference? Please check all the variables and coefficients are properly described!
Ans.These mistakes are corrected and κMS is using for Multi-wall hybrid nanopartilce.
Q4. The only section which has improved with respect to the original is section 6 "Results and Discussions". Sections 1-5 need to be thoroughly revised.
Ans. All the sections are revised. The corrections are made for the paper quality and your kind suggestions.

Round 3
Reviewer 1 Report
Please try to improve the quality of your manuscript (and responses) next time.